# Design and Implementation of an Active Vibration Control Algorithm Using Servo Actuator Control Installed in Series with a Spring-Damper

Soo-Min Kim , Dae W. Kim and Moon K. Kwak *

Department of Mechanical, Robotics and Energy Engineering, Dongguk University-Seoul, Seoul 04620, Republic of Korea
* Correspondence: kwakm@dgu.ac.kr

**Abstract:** The membrane-type air spring can be used to suppress lateral vibration of a vibration isolation table. However, compared to voice coil actuators, pneumatic actuators are difficult to use for precise vibration control, because servo valves have nonlinear dynamic characteristics. Therefore, actuators, such as voice coil actuators, can be placed in parallel with air springs, allowing force-type actuators to provide additional force to the system. These actuators generate force. In the case of a ball-screw mechanism device or a linear servomotor, it is an actuator that generates displacement. These actuators are represented as serial active systems. Serial active systems are structurally simpler than parallel active systems. However, there are very few studies on vibration isolation systems using serial active systems compared to parallel active systems. As the two are different types of systems, a new control algorithm suitable for the serial active system is needed. This study proposes a system in which an actuator capable of accurately controlling displacement is connected in series with a support spring-damper. A new active vibration control algorithm for the proposed control system is also developed, which is termed the position input and position output. The proposed control algorithm uses the displacement of the system as an input and outputs the desired displacement of the actuator installed in series with the damper and spring. The proposed control algorithm increases the damping at the target frequency and reduces the response of the system. Numerical studies and experiments were conducted on the single-degree-of-freedom and multi-degree-of-freedom systems. The results show the efficacy of the proposed control system and the novel control algorithm for the vibration suppression of the lateral vibration of a vibration isolation table.

**Keywords:** servo actuator; position input and position output (PIPO); serial active system; vibration isolation

## 1. Introduction

Vibration control is a crucial technology used in various structures such as buildings, bridges, vehicles, and machinery to reduce unwanted vibrations and noise [1]. Its main purpose is to maintain the stability and structural integrity of these structures while improving comfort, safety, and reducing the risk of damage to sensitive equipment. Additionally, vibration control can also enhance the efficiency and performance of many systems by reducing vibrations that can cause machines to wear out quickly, leading to the extension in the life of the equipment and a reduction in operational costs.

A vibration isolation table is used to minimize the transmission of vibrations from the environment to sensitive equipment or device. Vibration can cause issues in various applications, such as affecting delicate instruments and samples in laboratory settings or malfunctioning sensitive machinery in manufacturing facilities. Vibration isolation tables use special materials, such as rubber and foam, to separate the equipment from the vibrating surface below and decrease the amount of vibration transmitted to the equipment. They are used in laboratories, manufacturing, medical imaging, semiconductor fabrication, and

optical metrological applications to provide a stable and controlled environment, resulting in improved accuracy, quality, and reduced errors and malfunctions.

When designing vibration isolation tables, passive, semi-active, and active vibration control systems are used to suppress undesirable ground vibrations and disturbances caused by the table-mounted devices. In particular, this study addresses active vibration control for the lateral vibration of the vibration isolation table. As passive systems alone cannot suppress vibrations, active or semi-active systems are added to the vibration control of vibration isolation tables. However, not many actuators are available for vibration control. Depending on the type of actuator, anti-vibration systems can be classified into typical four types, as shown in Figure 1. Of course, there are other systems besides these systems.

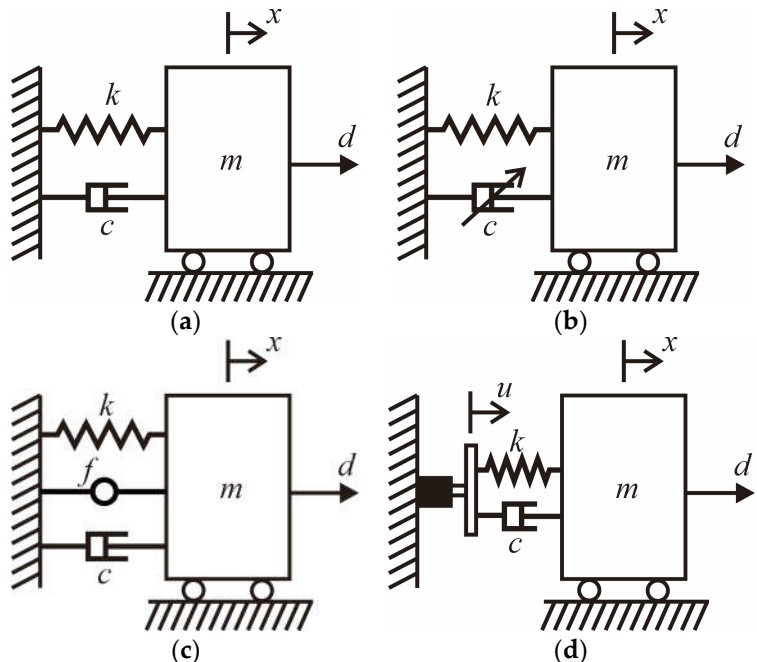

**Figure 1.** Vibration control systems. (**a**) Passive system; (**b**) Semi-active system; (**c**) Parallel active system; (**d**) Serial active system.

In this figure, $m$, $c$, $k$ are the mass, damping coefficient, and spring constant, respectively; $x$ is the displacement of the mass, $f$ is the actuator force, $u$ is the base motion, and $d$ is the disturbance acting on the mass.

Typically, in the passive vibration isolation system shown in Figure 1a, springs and damping elements are used to limit the amplitude of vibration and dissipate energy from the system [2,3]. Devices such as the tuned mass damper (TMD), tuned liquid damper (TLD), tuned liquid, column ball spring sliding damper (TLCBSSD), and tuned liquid column ball spring rolling damper (TLCBSRD) [4] can also be used. Passive vibration isolation devices are used to reduce the transmission of unwanted vibrations from a vibrating source to a sensitive structure. They can be highly effective in many applications, such as reducing noise and vibrations in sensitive equipment such as microscopes, laser systems, and precision machinery. Passive vibration isolation devices are generally less expensive than active vibration isolation systems, making them a cost-effective solution for many applications. Passive vibration isolation devices are easy to install and require no power or maintenance, making them a convenient solution for many applications. Passive vibration isolation devices do not consume any energy, making them an environmentally friendly solution. Passive vibration isolation devices have a long lifespan and require little to no maintenance, making them a reliable solution for many applications. However, there are disadvantages of passive vibration isolation devices. Passive vibration isolation devices may not provide the same level of isolation performance as active vibration isolation systems. Passive vibration isolation devices can be affected by changes in temperature,

humidity, and other environmental factors, which can affect their performance over time. Passive vibration isolation devices may not provide effective isolation over a wide range of frequencies, making them less suitable for applications with complex vibration spectra. Passive vibration isolation devices cannot be easily adjusted to accommodate changes in the vibrating source or the sensitive structure, making them less flexible than active vibration isolation systems. Passive systems have predetermined properties that cannot be adjusted while the system is operating. Although passive vibration isolation systems remain an effective solution to a wide range of vibration problems today, there are many applications where passive systems alone cannot satisfy customer requirements.

Semi-active systems use a variable damping technique, as shown in Figure 1b. Depending on the states, the variable damper changes their damping coefficient to minimize the displacement. Semi-active vibration isolation systems offer some of the benefits of both passive and active vibration isolation systems. Compared to passive vibration isolation systems, semi-active vibration isolation systems can provide improved isolation performance, especially in the presence of changing or unpredictable vibrations. Semi-active vibration isolation systems can be adjusted to accommodate changes in the vibrating source or the sensitive structure, making them more flexible than passive vibration isolation systems. Semi-active vibration isolation systems consume less energy than active vibration isolation systems, making them a more energy-efficient solution. Compared to active vibration isolation systems, semi-active vibration isolation systems are less expensive, making them a cost-effective solution for many applications. However, there are also disadvantages of using semi-active vibration isolation systems. Semi-active vibration isolation systems are more complex than passive vibration isolation systems, requiring specialized knowledge and technical expertise to design, install, and maintain. The performance of semi-active vibration isolation systems depends on the design of the control algorithms, which can be affected by changes in the environment, sensor noise, and other factors. Semi-active vibration isolation systems require regular maintenance to ensure proper functioning, which can increase their overall cost of ownership. Semi-active vibration isolation systems require a power source, which may not be available in some remote or harsh environments.

The most widely used variable damping device is the Magneto-Rheological Fluid (MRF) damper [5–8], and sky-hook control is the distinctive control algorithm employed by the MRF damper [9]. The sky-hook control is a very simple but effective vibration control algorithm that generates voltage according to the absolute and relative velocities of the main body. However, the sky-hook control algorithm requires the absolute and relative velocities that cannot be measured directly.

Active vibration control systems use actuators, sensors, and controllers to directly suppress vibration in the system [10]. Actuators must be able to provide the desired force or displacement to the system, depending on the type of actuator. Sensors are used to detect motion in systems involving acceleration, velocity, or relative displacement. The controller calculates the required external force or displacement using control algorithms and sends a signal to control the actuator. The controller can be either an analog circuit or a digital controller. Depending on the position of the actuator, there are two types of active control. Figure 1c shows an actuator installed parallel to the damper and spring, whereas Figure 1d shows an actuator installed in series with the damper and spring.

Popular actuators used to transmit force to the vibration isolation model shown in Figure 1c are air springs [11–17], and voice-coil actuators (VCA) [18–20]. The air springs are actually pneumatic actuators, so that they can support static loads while providing dynamic force. Air springs are commonly used in the suspension systems of vehicles, such as cars, trucks, buses, and trains, to provide a smooth and comfortable ride. They are used in place of traditional mechanical springs to provide improved ride quality, adjustability, and load-carrying capabilities. Air springs are also used in a wide range of industrial equipment, including machine tools, packaging machines, and conveyor systems, to isolate vibration and reduce shock transmission. Overall, air springs are versatile and flexible

components that can be used in a wide range of applications to provide improved vibration isolation and shock absorption.

A voice-coil actuator is a type of electromagnetic actuator that is commonly used in vibration control systems. It consists of a coil of wire that is placed within a magnetic field, and when a current is passed through the coil, a force is generated that can be used to control the position of an object. In vibration control systems, voice-coil actuators are used in active vibration isolation systems to control the motion of a sensitive structure, such as a delicate instrument or optical system. The actuator is attached to the sensitive structure, and its motion is actively controlled in response to incoming vibrations, effectively isolating the structure from the vibrating source. The effectiveness of the vibration control system depends on the accuracy of the control algorithm and the responsiveness of the voice-coil actuator. Voice-coil actuators are fast and highly responsive, making them a popular choice for many active vibration control applications.

When a VCA is used with an air spring, the air spring supports only the static load, whereas the VCA provides the dynamic force. Both pneumatic actuators and VCAs require a command voltage to generate force. In the case of a pneumatic actuator, the command voltage controls the opening of the servo valve to adjust the force, and in the case of a VCA, the command voltage is converted into current through a power amplifier to generate the electro-magnetic force. In comparison to the VCA, it seems difficult to produce accurate control force by using the pneumatic actuator because of nonlinearity.

Most active vibration control is performed in the form of Figure 1c, in which actuators capable of generating force are connected in parallel. Compared to the parallel active isolation system in Figure 1c, not much research has been carried out on the serial active isolation system in Figure 1d.

A control algorithm must be developed according to the type of sensor and actuator. The Positive Position Feedback (PPF) controller can be used when using position sensors and force-producing actuators [21]. The PPF controller has been widely used for structures equipped with piezoelectric sensors and actuators attached to structures [22–24]. However, accelerometers are the most widely used sensors for vibration measurement that actually measure absolute acceleration. The so-called Virtual Tuned Mass Damper (VTMD) control was developed to utilize an accelerometer as a sensor, and use an inertial actuator [25]. In applying the VTMD control, the displacement of the inertial actuator should be accurately controlled. If the absolute position and absolute velocities can be measured, a full-state feedback control can be designed, such as a linear quadratic regulator (LQR). Because it is impossible to measure absolute position and absolute velocity, linear quadratic Gaussian (LQG) control is often resorted to, which is actually a Kalman filter. Yang et al. [26] proposed a negative acceleration feedback (NAF) control algorithm for an active tuned mass damper (ATMD) in a situation where the displacement of the proof mass is accurately controlled using an AC servo motor, and the accelerometer signal is measured.

The air spring and VCA described above are actuators that generate force. A ball-screw mechanism can be used to generate the desired displacement. However, displacement generating actuators cannot be installed parallel to spring dampers because the ball-screw mechanism acts as a lock when it is not working. In other words, displacement type actuator cannot be used in parallel active type. So, in this case, the case of connecting the linear actuator to the spring-damper in series has to be considered, as shown in Figure 1d. The serial type shown in Figure 1d is structurally simpler in reality than the parallel type shown in Figure 1c. However, there are very few studies on vibration isolation systems using serial active systems compared to parallel active systems. As the two are different types of systems, a new control algorithm suitable for the serial active system is also needed.

In this study, we are interested in the control of lateral vibrations of the vibration isolation table and assume that the displacement of the table can be measured, and the displacement of the base plate (see Figure 1d) can be accurately controlled. In addition, new control algorithms so called position-input and position-output (PIPO) controls have

been developed to cope with the mentioned problems. Theoretical and experimental result show the efficacy of the proposed serial-type actuator and the control algorithm.

## 2. Single-Degree-of-Freedom Control Design

Consider the single-degree-of-freedom (SDOF) subjected to the actuator installed in series with a damper and a spring, as shown in Figure 1d. The equation of motion can be written as:

$$m\ddot{x} + c\dot{x} + kx = c\dot{u} + ku + d \tag{1}$$

where $\cdot = d/dt$ represents the time-derivative. Dividing Equation (1) by $m$ obtain:

$$\ddot{x} + 2\zeta\omega_n\dot{x} + \omega_n^2 x = 2\zeta\omega_n\dot{u} + \omega_n^2 u + \omega_n^2\bar{d} \tag{2}$$

where:

$$\omega_n = \sqrt{\frac{k}{m}},\ \zeta = \frac{c}{2m\omega_n},\ \bar{d} = \frac{d}{k} \tag{3}$$

In contrast to general equations of motion subjected to force and disturbance, the right-hand side of Equation (2) consists of the displacement and velocity of the displacement actuator. Hence, the active vibration control algorithms introduced earlier that produce the desired force cannot be directly applied to Equation (2).

In this study, we propose the following control algorithm for the system given by Equation (2).

$$\ddot{u} + 2\zeta_f\omega_n\dot{u} + \omega_n^2 u = g\left(2\zeta_f\omega_n\dot{x} + \omega_n^2 x\right) \tag{4}$$

The control algorithm given by Equation (4) is the reciprocal of Equation (2) except for the gain and damping factor. The transfer function of the compensator given by Equation (4) can be also expressed as:

$$\frac{U(s)}{X(s)} = H(s) = \frac{g\left(2\zeta_f\omega_n s + \omega_n^2\right)}{s^2 + 2\zeta_f\omega_n s + \omega_n^2} \tag{5}$$

where $X(s) = \mathcal{L}\{x(t)\}$, $U(s) = \mathcal{L}\{u(t)\}$, and $\mathcal{L}$ is the Laplace operator. The compensator given by Equation (5) will be referred to in this paper as the PIPO controller. Unlike the PPF controller, the output of the PIPO is the desired displacement of the actuator installed in series with the damper and the spring. The novelty of the proposed single-input and single-output (SISO) PIPO controller given by Equation (5) is that its form is very simple and can be easily implemented using analog circuits or digital controllers. Of course, a sensor capable of precisely measuring the displacement of the table and an actuator capable of accurately realizing the desired displacement are required. The smaller the damping factor of the compensator, the greater the damping effect, but the narrower the controllable frequency band.

Figure 2 shows the Bode diagram of the PIPO controller for various $\zeta_f$, where $\omega_n = 1$ rad/s, $g = 0.1$ are used. The magnitude plot shown in Figure 2a shows the typical second-order low-pass filter, whereas the phase plot shown in Figure 2b indicates that the phase at the filter frequency is 90 degree, which implies active damping. Hence, it can be said that the control algorithm given by Equation (5) can affect the system by increasing the damping at the target frequency. It is desirable to give the proposed control algorithm a larger frequency band. So, it seems preferable to use $\zeta_f = 0.3$ as in the case of the PPF controller.

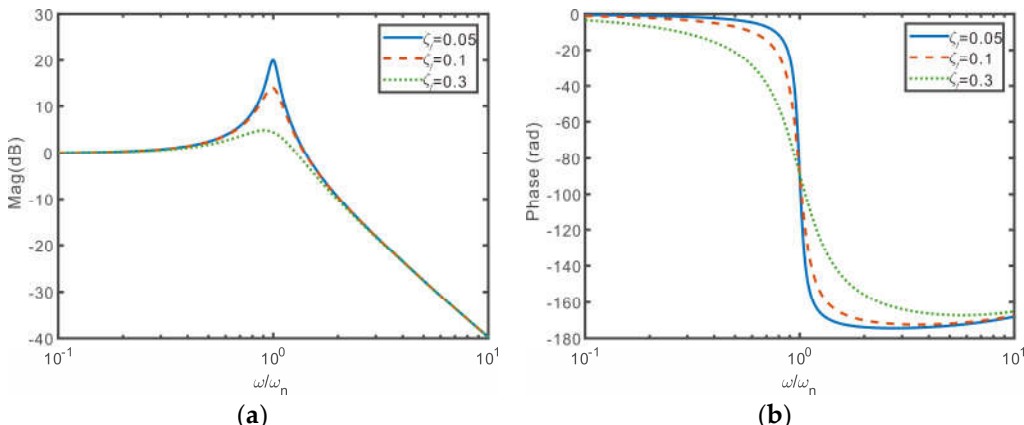

**Figure 2.** Bode diagram for the proposed compensator. (**a**) Magnitude; (**b**) Phase.

As stated above, the compensator transfer function has 90-degree shift at the filter frequency, so that it can create active damping for the addressed problem. Combining Equation (2) with Equation (4) results in the following closed-loop matrix equation of motion.

$$\begin{Bmatrix} \ddot{x} \\ \ddot{u} \end{Bmatrix} + \begin{bmatrix} 2\zeta\omega_n & -2\zeta\omega_n \\ -2g\zeta_f\omega_n & 2\zeta_f\omega_n \end{bmatrix} \begin{Bmatrix} \dot{x} \\ \dot{u} \end{Bmatrix} + \begin{bmatrix} \omega_n^2 & -\omega_n^2 \\ -g\omega_n^2 & \omega_n^2 \end{bmatrix} \begin{Bmatrix} x \\ u \end{Bmatrix} = \begin{bmatrix} \omega_n^2 \\ 0 \end{bmatrix} \overline{d} \tag{6}$$

Applying the Laplace transform to Equation (6) obtains:

$$\begin{bmatrix} s^2 + 2\zeta\omega_n s + \omega_n^2 & -(2\zeta\omega_n s + \omega_n^2) \\ -g\left(2\zeta_f\omega_n s + \omega_n^2\right) & s^2 + 2\zeta_f\omega_n s + \omega_n^2 \end{bmatrix} \begin{Bmatrix} X(s) \\ U(s) \end{Bmatrix} = \begin{bmatrix} \omega_n^2 \\ 0 \end{bmatrix} \overline{D}(s) \tag{7}$$

where $\overline{D}(s) = \mathcal{L}\left\{\overline{d}(t)\right\}$. Then, the determinant of the matrix becomes:

$$det = s^4 + 2(\zeta + \zeta_f)\omega_n s^3 + 2\left[1 + 2(1-g)\zeta\zeta_f\right]\omega_n^2 s^2 + 2(1-g)(\zeta + \zeta_f)\omega_n^3 s + (1-g)\omega_n^4 \tag{8}$$

Applying the Routh–Hurwitz stability criteria [27] to Equation (8), the following stability condition can be obtained:

$$\text{Stable if } 0 < g < 1 \tag{9}$$

Such as the PPF controller, the stability condition is static, which means that the stability condition does not depend on frequency, which is a very desirable stability condition from the point of view of controller design. The closed-loop system, Equation (6), can be drawn by the block diagram shown in Figure 3.

Figure 4 shows the Bode diagram of the closed-loop transfer function for various gains, where $\zeta_f = 0.3$ was employed. It can be seen from Figure 4 that $g = 0.1$ is good enough, and a large gain rather degrades the performance of the closed loop system. Figure 5 shows the uncontrolled and controlled impulse responses when $\zeta = 0.01$, $\zeta_f = 0.3$, $g = 0.1$, $\omega_n = 20.1$ rad/s. Using logarithmic decrement, the damping factor of the controlled system was found to be 0.107. Figure 6 shows the displacement response the controller is turned on at 4 s. As shown in Figures 5 and 6, it can be seen that the PIPO controller proposed in this study can reduce the response very effectively.

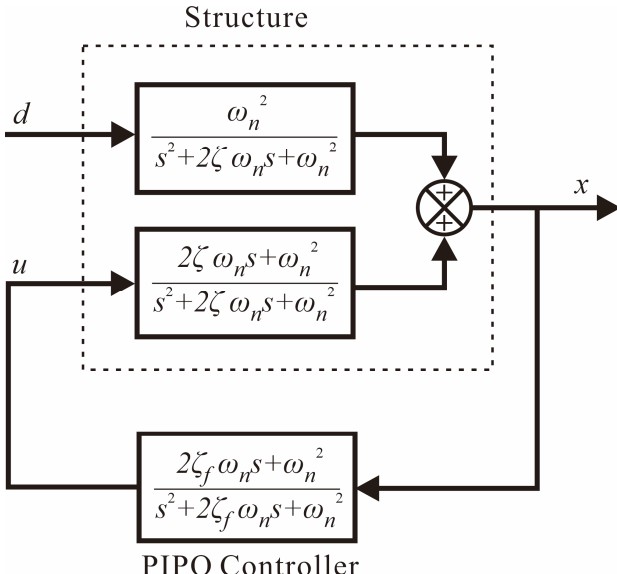

**Figure 3.** Block diagram of the closed-loop system.

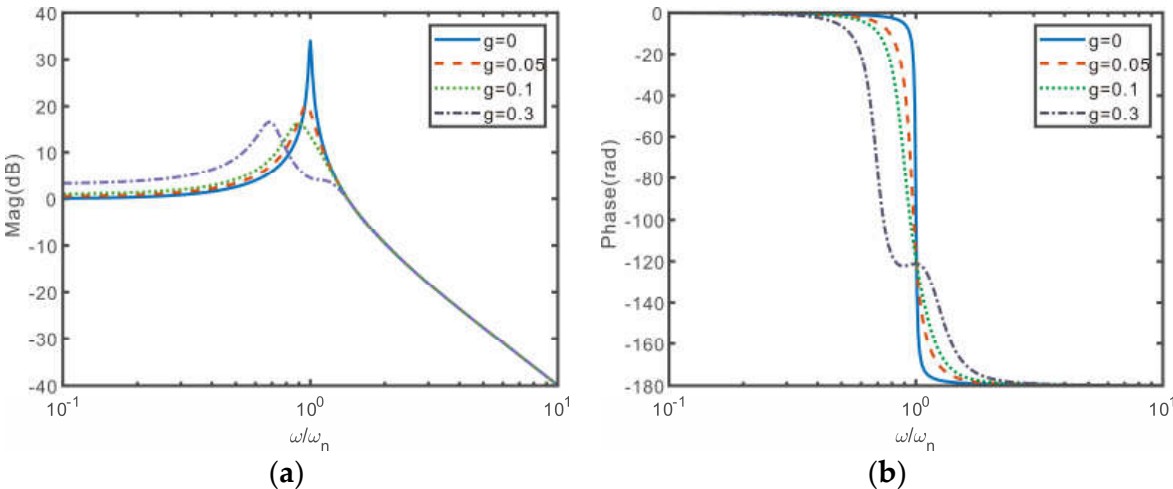

**Figure 4.** The Bode diagram of the closed-loop system. (**a**) Magnitude; (**b**) Phase.

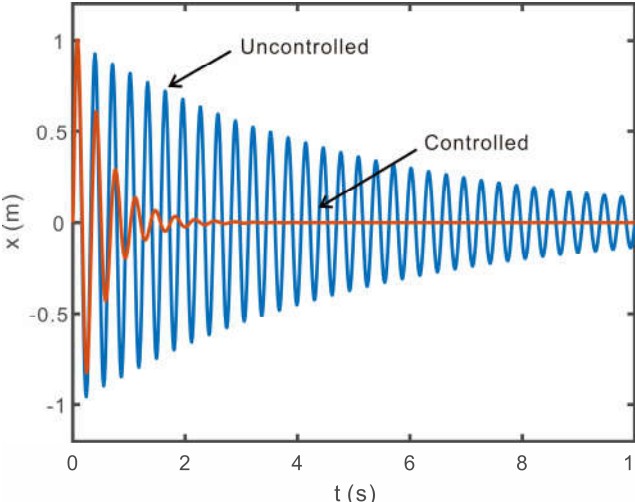

**Figure 5.** Free vibration simulation.

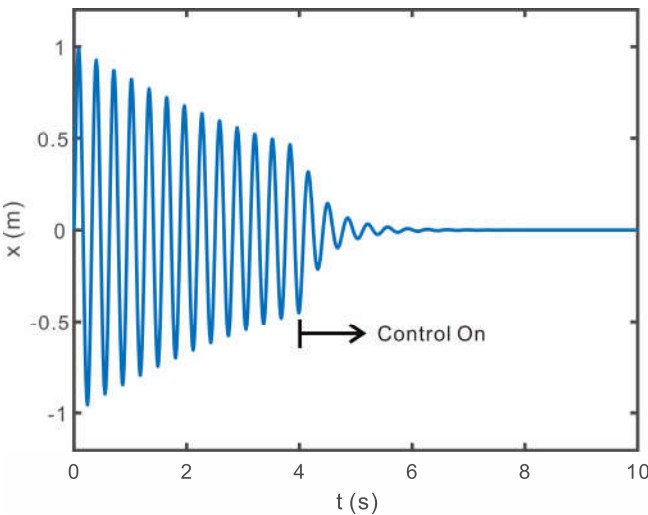

**Figure 6.** The time history of displacement.

## 3. Experiments for the SDOF System

Figure 7 shows the simple model that was set up to prove the validity of the proposed PIPO controller given by Equation (5), whereas Figure 8 shows the wire connection diagram for the experiment. The displacement of the floor was measured using the laser displacement sensor, (M3L/20), from MEL, and the base movement was made by the linear servo motor, (LM-H3P2A-07P-BSS0), Mitsubishi Electric Co., Tokyo, Japan, whereas the PIPO control algorithm was implemented using dSPACE MicroLabBox.

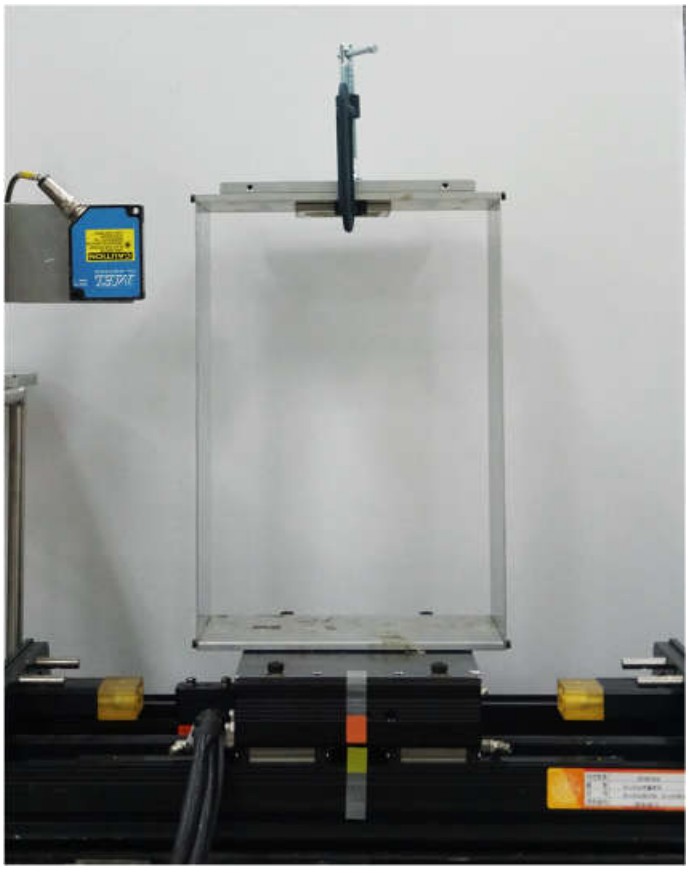

**Figure 7.** Experimental testbed.

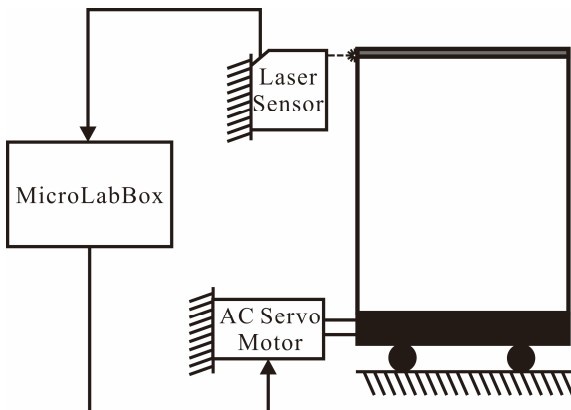

**Figure 8.** Wiring diagram.

Through the free vibration experiment, the fundamental frequency of the model was found to be 3.2 Hz. The PIPO controller is implemented as a Simulink block diagram, as shown in Figure 9. The voltage signal output from the laser sensor is converted into actual displacement by multiplying the gain corresponding to the sensor characteristic. A moving average filter is used to remove sensor noise, and the displacement is used as an input to the PIPO controller. The outputted control displacement from PIPO controller is input to the servo motor. The proportional-integral-derivative (PID) controller is used for accurate displacement-tracking control of a linear servo motor [28]. For real-time displacement tracking, the servo motor driver's mode was set to speed mode. For details, refer to reference [25]. The gain values of the PIPO controller and the PID were experimentally set. When a control signal is suddenly input to the servo motor, the motor is shocked, so a switch was created in the program so that the control signal was input smoothly. Figure 10 shows the free vibration response of the uncontrolled system and the controlled system, whereas Figure 10 shows how quickly vibrations are suppressed when the PIPO controller is turned on. Using logarithmic decrement, the damping factor of the uncontrolled system is calculated as 0.033, and the damping factor of the controlled system is 0.168. Figures 10 and 11, the response curves obtained through experimentation, are very similar to the theoretical response curve shown in Figures 5 and 6. Therefore, it can be confirmed that the performance of the PIPO controller predicted by the theory in the previous section is valid. As shown in Figures 9–11, the effectiveness of the proposed PIPO controller has been demonstrated and proved experimentally. The reason that the very small residual vibration remains in the control state is that the input voltage is less than the minimum input voltage of the servo motor; that is, it corresponds to the dead band of the servo motor. Therefore, the phenomenon occurred due to the characteristics of the linear servo motor, not the PIPO controller.

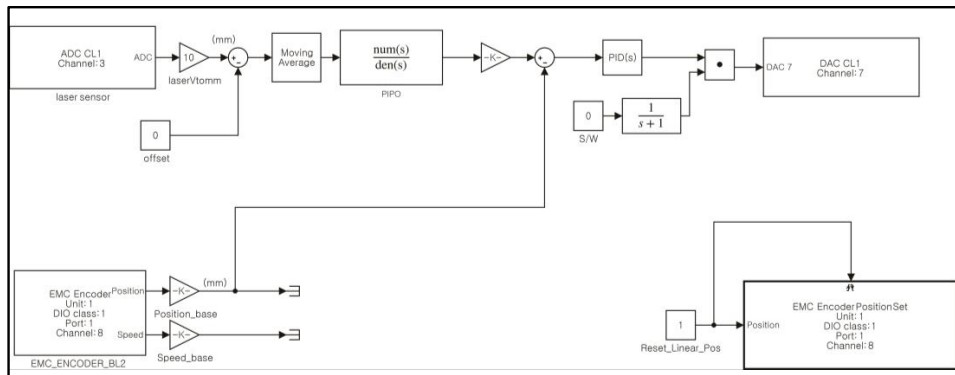

**Figure 9.** Simulink block diagram for the PIPO controller.

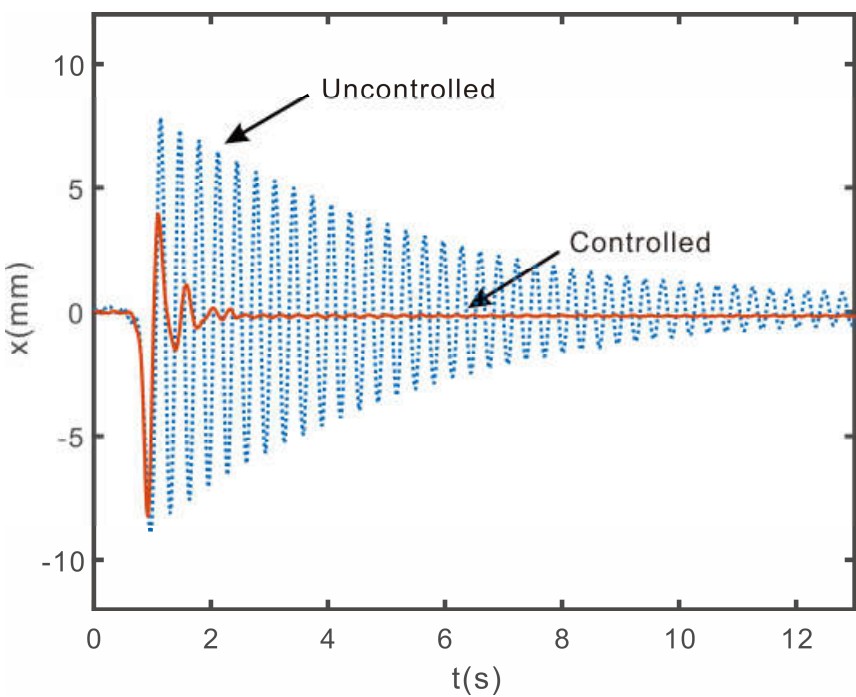

**Figure 10.** Free vibration experiment.

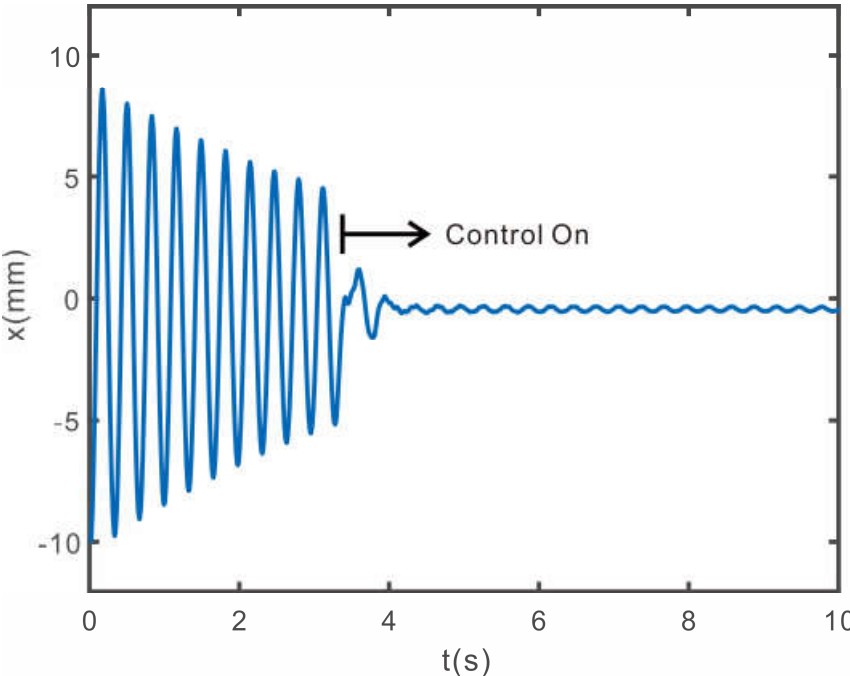

**Figure 11.** Free vibration response when the controller is turned on in the middle of the time.

## 4. Multi-Input Multi-Output Control Design

Consider the multi-degree-of-freedom (MDOF) model shown in Figure 12. Our control goal remains to suppress vibrations of the first floor.

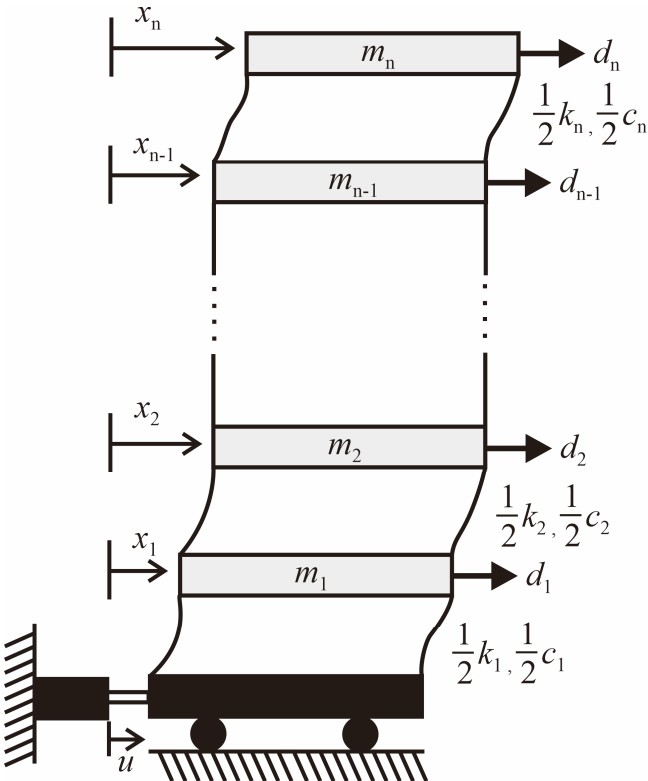

**Figure 12.** Multi-degree-of-freedom model.

The matrix equations of motion can be expressed as:

$$\mathbf{M}\ddot{\mathbf{x}} + \mathbf{C}\dot{\mathbf{x}} + \mathbf{K}\mathbf{x} = \mathbf{B}_u\mathbf{u} + \mathbf{d} \tag{10}$$

where $\mathbf{x} = [x_1\ x_2\ \cdots\ x_n]^{\mathrm{T}}$, $\mathbf{u} = [\,u\ \dot{u}\,]^{\mathrm{T}}$, $\mathbf{d} = [d_1\ d_2\ \cdots\ d_n]^{\mathrm{T}}$, and:

$$\mathbf{M} = \begin{bmatrix} m_1 & 0 & \cdots & 0 \\ 0 & m_2 & \cdots & 0 \\ \vdots & \vdots & \ddots & \vdots \\ 0 & 0 & \cdots & m_n \end{bmatrix}, \ \mathbf{C} = \begin{bmatrix} c_1+c_2 & -c_2 & \cdots & 0 \\ -c_2 & c_2+c_3 & \cdots & 0 \\ \vdots & \vdots & \ddots & \vdots \\ 0 & 0 & \cdots & c_n \end{bmatrix},$$

$$\mathbf{K} = \begin{bmatrix} k_1+k_2 & -k_2 & \cdots & 0 \\ -k_2 & k_2+k_3 & \cdots & 0 \\ \vdots & \vdots & \ddots & \vdots \\ 0 & 0 & \cdots & k_n \end{bmatrix}, \ \mathbf{B}_u = \begin{bmatrix} k_1 & c_1 \\ 0 & 0 \\ \vdots & \vdots \\ 0 & 0 \end{bmatrix} \tag{11}$$

Equation (10) can be rewritten as a first-order matrix differential equation:

$$\dot{\mathbf{z}} = \mathbf{A}_{xx}\mathbf{z} + \mathbf{A}_{xu}\mathbf{u} + \mathbf{B}_d\mathbf{d} \tag{12}$$

where, $\mathbf{z} = [\,\mathbf{x}^{\mathrm{T}}\ \dot{\mathbf{x}}^{\mathrm{T}}\,]^{\mathrm{T}}$, and:

$$\mathbf{A}_{xx} = \begin{bmatrix} \mathbf{0} & \mathbf{I} \\ -\mathbf{M}^{-1}\mathbf{K} & -\mathbf{M}^{-1}\mathbf{C} \end{bmatrix}, \ \mathbf{A}_{xu} = \begin{bmatrix} \mathbf{0} \\ \mathbf{M}^{-1}\mathbf{B}_u \end{bmatrix}, \ \mathbf{B}_d = \begin{bmatrix} \mathbf{0} \\ \mathbf{M}^{-1} \end{bmatrix} \tag{13}$$

We design each PIPO controller as follows to suppress $m \ (\leq n)$ natural modes:

$$
\begin{aligned}
\ddot{u}_1 + 2\zeta_f\omega_1\dot{u}_1 + \omega_1^2 u_1 &= g_1\left(2\zeta_f\omega_1\dot{x}_1 + \omega_1^2 x_1\right) \\
\ddot{u}_2 + 2\zeta_f\omega_2\dot{u}_2 + \omega_2^2 u_2 &= g_2\left(2\zeta_f\omega_2\dot{x}_1 + \omega_2^2 x_1\right) \\
&\vdots \\
\ddot{u}_m + 2\zeta_f\omega_m\dot{u}_m + \omega_m^2 u_m &= g_m\left(2\zeta_f\omega_m\dot{x}_1 + \omega_m^2 x_1\right)
\end{aligned}
\tag{14}
$$

where, $g_i(i = 1, 2, \dots, m)$ is the control gain for $i$th natural mode. In the real system, it is not possible to install sensors that measure all displacements of the mass. Models with high degrees of freedom can cause physical or financial challenges. Therefore, we assume in this study that only the first-floor displacement can be measured. Equation (14) can be rewritten as a matrix equation:

$$
\ddot{\mathbf{u}}_m + 2\zeta_f\mathbf{\Omega}_m\dot{\mathbf{u}}_m + \mathbf{\Lambda}_m\mathbf{u}_m = 2\zeta_f\mathbf{G}_\omega\mathbf{E}_1\dot{\mathbf{x}} + \mathbf{G}_\lambda\mathbf{E}_1\mathbf{x}
\tag{15}
$$

where, $\mathbf{u}_m = \begin{bmatrix} u_1 \ u_2 \ \cdots \ u_m \end{bmatrix}^{\mathrm{T}}$, and:

$$
\mathbf{\Omega}_m = \begin{bmatrix} \omega_1 & 0 & \cdots & 0 \\ 0 & \omega_2 & \cdots & 0 \\ \vdots & \vdots & \ddots & \vdots \\ 0 & 0 & \cdots & \omega_m \end{bmatrix}, \ \mathbf{\Lambda}_m = \begin{bmatrix} \omega_1^2 & 0 & \cdots & 0 \\ 0 & \omega_2^2 & \cdots & 0 \\ \vdots & \vdots & \ddots & \vdots \\ 0 & 0 & \cdots & \omega_m^2 \end{bmatrix},
$$

$$
\mathbf{G}_\omega = \begin{bmatrix} g_1\omega_1 \\ g_2\omega_2 \\ \vdots \\ g_m\omega_m \end{bmatrix}, \ \mathbf{G}_\lambda = \begin{bmatrix} g_1\omega_1^2 \\ g_2\omega_2^2 \\ \vdots \\ g_m\omega_m^2 \end{bmatrix}, \ \mathbf{E}_1 = \begin{bmatrix} 1 \ 0 \ \cdots \ 0 \end{bmatrix}
\tag{16}
$$

In reality, Equation (15) represents a single-input and multi-output (SIMO) controller, which can also be converted to a first-order matrix differential equation:

$$
\dot{\mathbf{v}} = \mathbf{A}_{uu}\mathbf{v} + \mathbf{A}_{ux}\mathbf{z}
\tag{17}
$$

where, $\mathbf{v} = \begin{bmatrix} \mathbf{u}_m^{\mathrm{T}} \ \dot{\mathbf{u}}_m^{\mathrm{T}} \end{bmatrix}^{\mathrm{T}}$, and:

$$
\mathbf{A}_{uu} = \begin{bmatrix} \mathbf{0} & \mathbf{I} \\ -\mathbf{\Lambda}_m & -2\zeta_f\mathbf{\Omega}_m \end{bmatrix}, \ \mathbf{A}_{ux} = \begin{bmatrix} \mathbf{0} & \mathbf{0} \\ \mathbf{G}_\lambda\mathbf{E}_1 & 2\zeta_f\mathbf{G}_\omega\mathbf{E}_1 \end{bmatrix}
\tag{18}
$$

Because the sum of the outputs of each PIPO controller becomes the actual displacement of the base, this can be expressed as:

$$
u = u_1 + u_2 + \dots + u_n
\tag{19}
$$

Hence, we may write:

$$
\mathbf{u} = \mathbf{E}_m\mathbf{v}
\tag{20}
$$

where:

$$
\mathbf{E}_m = \begin{bmatrix} 1 & 1 & \cdots & 1 & 0 & 0 & \cdots & 0 \\ 0 & 0 & \cdots & 0 & 1 & 1 & \cdots & 1 \end{bmatrix}
\tag{21}
$$

Inserting Equation (20) into Equation (12) yields:

$$
\dot{\mathbf{z}} = \mathbf{A}_{xx}\mathbf{z} + \mathbf{A}_{xu}\mathbf{E}_m\mathbf{v} + \mathbf{B}_d\mathbf{d}
\tag{22}
$$

Combining Equations (17) and (22), the closed-loop state space equation can be written as:

$$\dot{\mathbf{p}} = \mathbf{A}_c\mathbf{p} + \mathbf{B}_c\mathbf{d}, \ y = x_1 = \mathbf{C}_c\mathbf{p} + \mathbf{D}_c\mathbf{d} \tag{23}$$

where $\mathbf{p} = [\ \mathbf{z}^T\ \mathbf{v}^T\ ]^T$, and:

$$\mathbf{A}_c = \begin{bmatrix} \mathbf{A}_{xx} & \mathbf{A}_{xu}\mathbf{E}_m \\ \mathbf{A}_{ux} & \mathbf{A}_{uu} \end{bmatrix}, \ \mathbf{B}_c = \begin{bmatrix} \mathbf{B}_d \\ \mathbf{0} \end{bmatrix}, \ \mathbf{C}_c = [1\ 0\ 0\ \cdots\ 0\ ], \ \mathbf{D}_c = [\mathbf{0}] \tag{24}$$

For the closed-loop state space equation given by Equation (23) to be stable, the real parts of the eigenvalues of matrix $\mathbf{A}_c$ must be negative, which must be checked before applying the controller.

## 5. Numerical Example for the MDOF System

Consider a three-degree-of-freedom model for numerical studies, where we are going to suppress the two lowest natural modes. So, in this case, $n = 3$, $m = 2$. The masses, spring constants, damping coefficients, and gains are as follows:

$$m_1 = m_2 = m_3 = 1\,\text{kg},\ k_1 = k_2 = k_3 = 100\,\text{N/m}$$
$$c_1 = c_2 = c_3 = 0.1\,\text{Ns/m},\ g_1 = g_2 = 0.1$$

The damping factor of the PIPO controller is set to $\zeta_f = 0.3$. Figure 13 shows magnitude and phase plots for this problem, where the peak amplitudes decrease as expected. Therefore, it can be said that the proposed SIMO PIPO controller is effective, even for the MIMO system. Figure 14 shows the uncontrolled and controlled impulse responses. Using logarithmic decrement, the damping factor of the uncontrolled system was calculated as 0.007, and the damping factor of the closed-loop controlled system was calculated as 0.068. Figure 15 shows the uncontrolled and controlled impulse responses, and that the MIMO PIPO controller can suppress vibrations effectively.

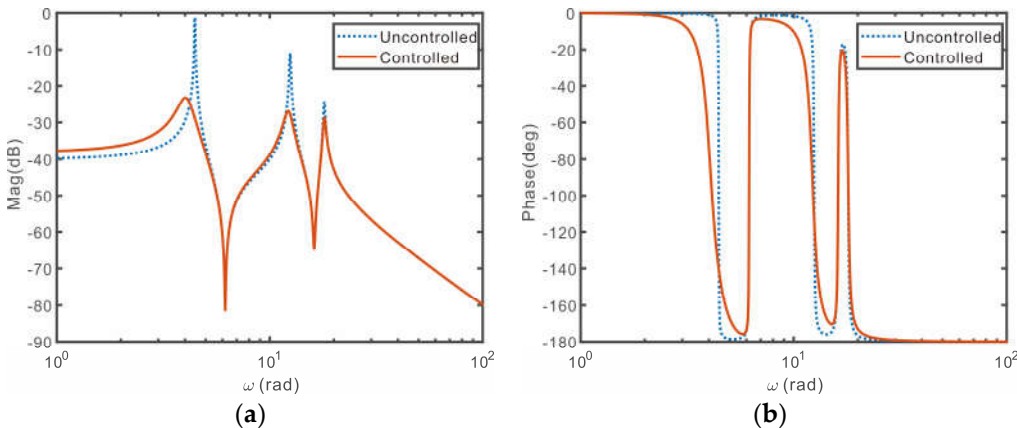

**Figure 13.** The bode diagram of the closed-loop system. (**a**) Magnitude; (**b**) Phase.

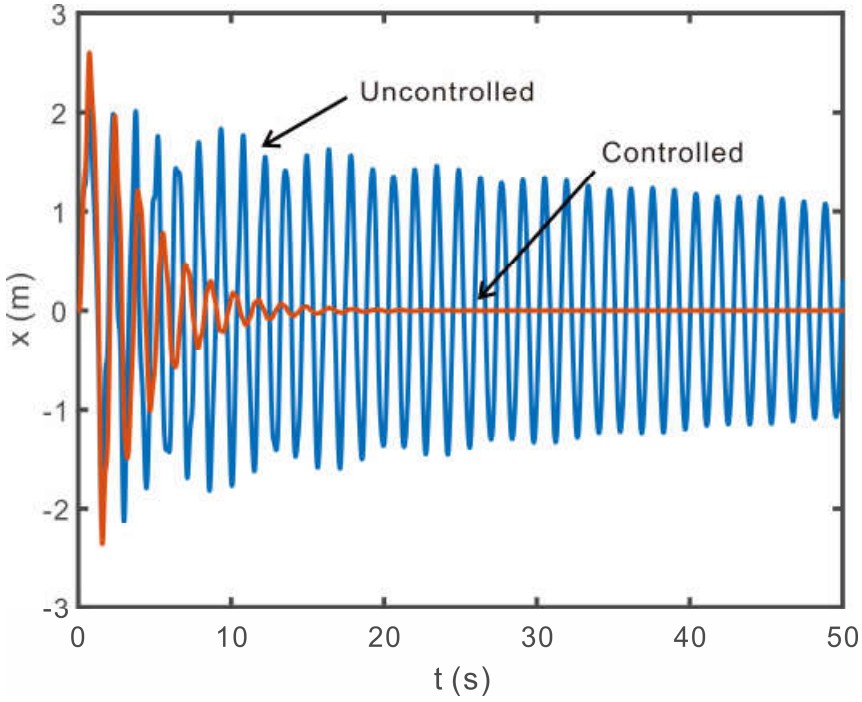

**Figure 14.** Free vibration experiment (the first mode excited).

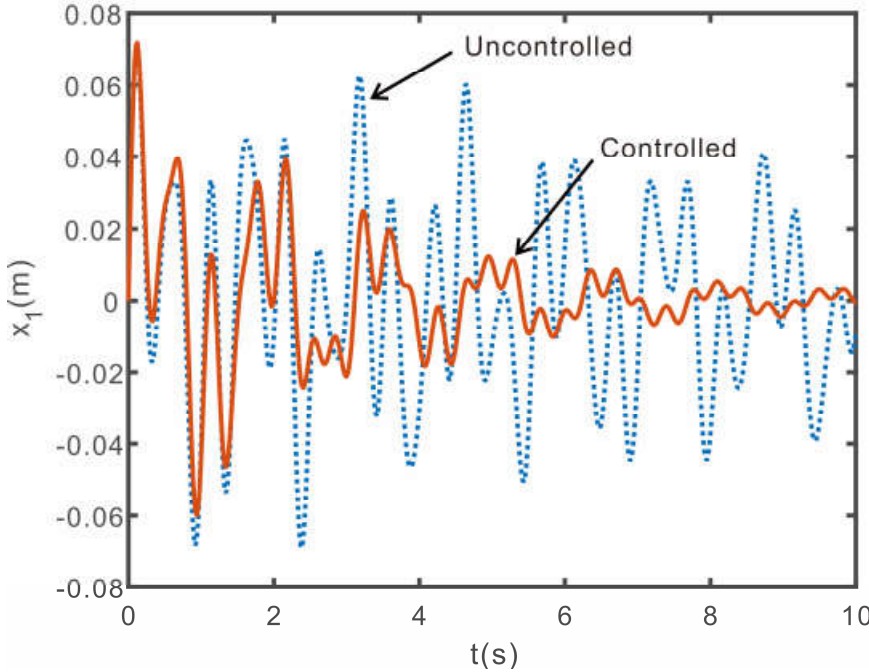

**Figure 15.** Impulse time-response of the closed-loop system.

## 6. Experiments for the MDOF System

Figure 16 shows the simple two-story building that was built to prove the effectiveness of the SIMO PIPO controller. The equipment used for vibration control of the single-degree-of-freedom system was used as was. Through the free vibration experiment, it was found that the natural frequencies were (1.3 and 5.3) Hz, respectively. So, as shown in the block diagram of Figure 17, a control algorithm combining two PIPO controllers for the first and the second natural frequencies in parallel was used. The block diagram in Figure 17 was also implemented as a Simulink program, as in Figure 18. Figure 19 shows the vibration response of the first floor when the first natural mode is excited, when there is

no controller, and when the MIMO PIPO controller is applied. Using logarithmic decrement, the damping factor of the uncontrolled system is calculated as 0.021, and the damping factor of the controlled system is calculated as 0.111. Figure 20 shows the vibration response of the first floor when the second natural mode is excited, when there is no controller, and when the MIMO PIPO controller is applied. Figures 19 and 20 confirm that the proposed SIMO PIPO controller proposed in this study is effective.

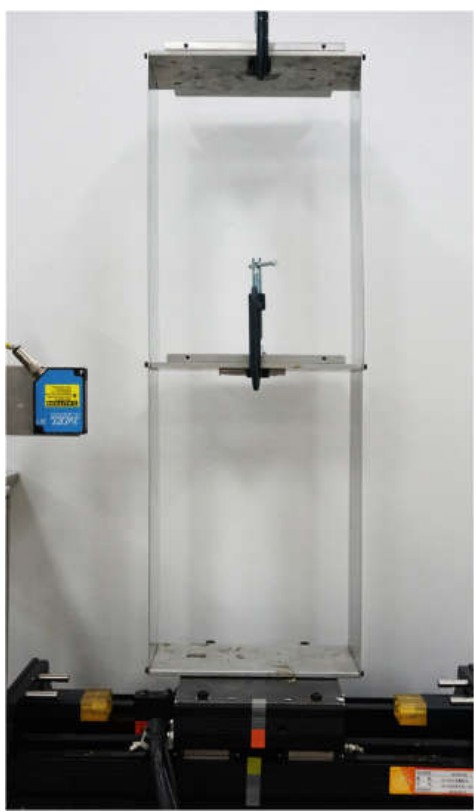

**Figure 16.** Experimental two-story building model.

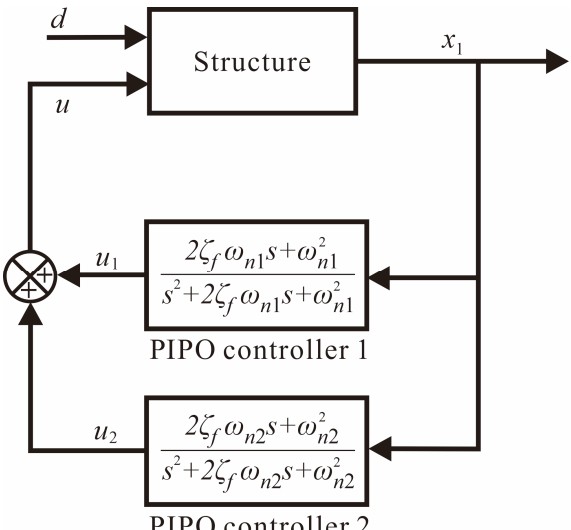

**Figure 17.** Block diagram for the closed-loop system.

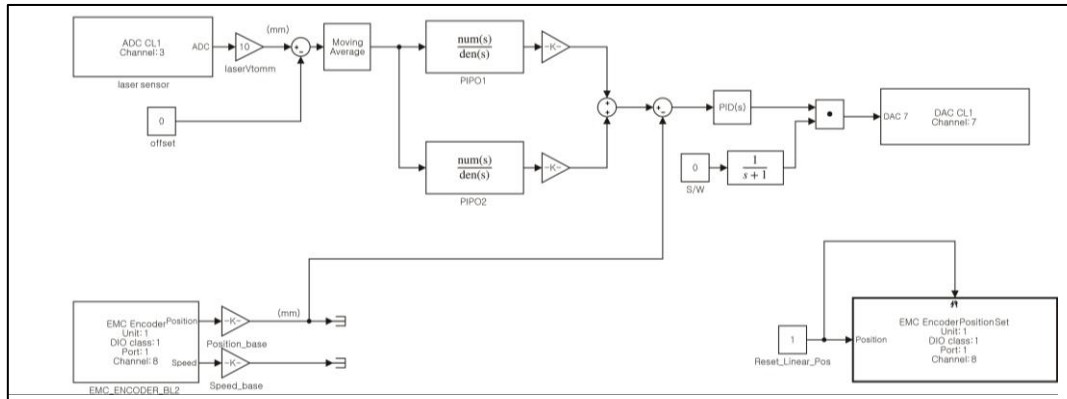

**Figure 18.** Simulink block diagram.

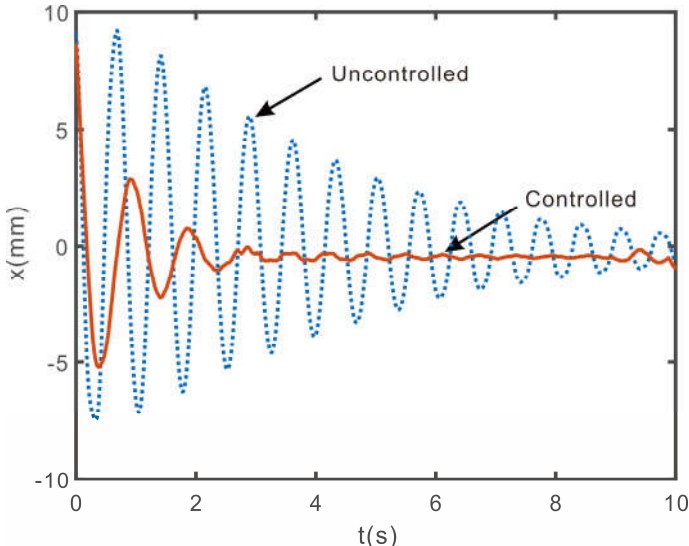

**Figure 19.** Free vibration experiment (the first mode excited).

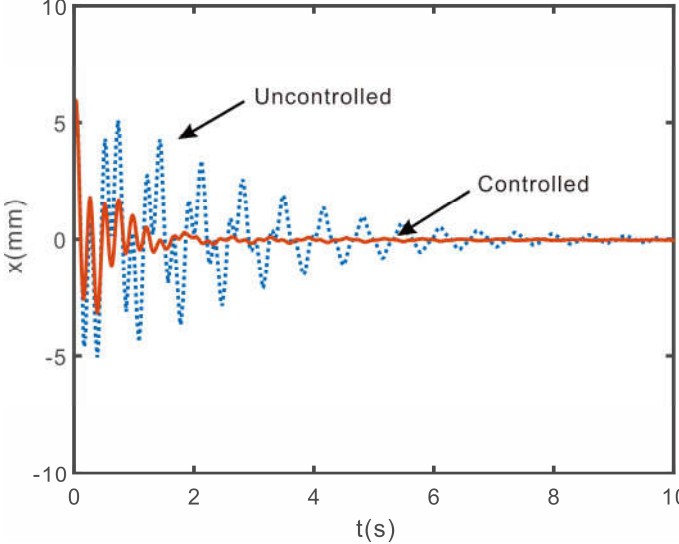

**Figure 20.** Free vibration experiment (the second mode excited).

## 7. Discussion and Conclusions

In this study, we propose a new vibration control system that can suppress lateral vibrations of the vibration isolation table. Most studies on active vibration isolation systems so far have been conducted on parallel active systems using air-spring or VCA. These generate force. There are also actuators that generate displacement. A vibration isolation system using these actuators can be expressed as a serial active system such as connecting a linear actuator and a spring-damper in series. As the parallel and serial active system are different system, a suitable control algorithm for each active system is required. However, there are very few studies on vibration isolation systems using serial active systems compared to parallel active systems. Therefore, in this study, a new algorithm for serial active type vibration isolation system was developed. The vibration suppression control system consisting of a support spring damper and an actuator that can accurately control displacement, where the actuator is connected in series with the spring damper.

The proposed control algorithm so-called position input and position output (PIPO) was proposed to create active damping. The PIPO control algorithm uses the displacement of the system as an input and outputs the desired displacement of the actuator. In this study, we first attempted to theoretically verify the effectiveness of the single-input and single-output (SISO) PIPO algorithm for a single degree-of-freedom (SDOF) model before experimentation. And the validity of the proposed control algorithm was investigated through simulation. In addition, to investigate the practical applicability of the controller confirmed through simulation, an experiment was conducted with a single-story building model connected to a linear servo motor.

The single-input and multi-output (SIMO) PIPO algorithm for the multi-degree-of-freedom (MDOF) system was also proposed in this study. Using the SIMO PIPO algorithm, a numerical simulation was conducted to check the vibration suppression of the two lowest natural modes of the three-degree-of-freedom model with only one displacement sensor. The simulation results show that the proposed SIMO PIPO control algorithm is effective in controlling many modes with a single actuator. In the experiment, a PIPO controller was designed using the Simulink program to control the first and second natural modes of a two-story building model. As a result of the experiment, both the first and second natural modes of the building model were suppressed quickly and reliably.

The proposed PIPO control system is particularly effective in controlling the horizontal vibration of the vibration isolation table because the implementation of the actuator connected in series with the air spring. With this control system, vibration suppression of the lateral vibration of the vibration isolation table can be achieved by connecting a simple displacement actuator to a membrane-type air spring in series.

The following conclusions were drawn from this study:

–   Air spring and VCA commonly used in vibration isolation tables, is a type of actuator that generates force. These are parallel active type. An actuator that generates displacement, such as a ball-screw mechanism device or a linear motor, can be expressed as a serial active system. However, there are very few studies on vibration isolation systems using serial active systems compared to parallel active systems.

–   The proposed PIPO control algorithm uses the displacement of the system as an input and outputs the desired displacement of the actuator installed in series with the damper and spring. The proposed configuration of the control system enables us to suppress lateral vibrations of the vibration isolation table accurately.

–   The stability condition of the proposed PIPO control algorithm is static. Hence, it does not depend on frequency and characteristics of system to be controlled. The stability of the PIPO control system can be easily guaranteed by adjusting only the gain.

–   The form of the proposed PIPO control algorithm has the form of a second-order low-pass filter and has a phase of 90-degree at the filter frequency. Therefore, active damping can be created while minimizing spillover to other modes. The proposed PIPO controller can be implemented by using either an analog circuit or a digital controller.

- The theoretically proven PIPO control algorithm for a single-degree-of-freedom system can be easily extended to a multi-degree-of-freedom system, thus simultaneously suppressing many natural modes.
- The SIMO PIPO control algorithm can control many natural modes with a single displacement sensor. However, the stability condition of the SIMO PIPO control algorithm needs to be checked before practical application.
- The efficacy of the PIPO control algorithms is proved experimentally using a single-degree-of-freedom model and a two-degree-of-freedom model.
- Using logarithmic decrement, the damping factors of the uncontrolled and controlled SDOF system were obtained. Numerically, we can increase the damping factor from 0.010 to 0.107 for the SDOF system. In the case of the experimental SDOF system, we can increase the damping factor from 0.033 to 0.168. In the case of the numerical MDOF system, we can increase the damping factor from 0.007 to 0.068. In the experiment, we can increase the damping factor of the MDOF system from 0.021 to 0.111. Both numerical and experimental results show the efficiency of the proposed control system and PIPO algorithm.
- The serial active type is structurally simpler than the parallel active type.
- If the proposed control system is applied to a vibration isolation table, it is possible to control vibrations that were difficult to precisely control with pneumatic control relatively easily.
- However, compared to the parallel active system, the serial active system must withstand the static load when installed in the vertical direction.

**Author Contributions:** Conceptualization, M.K.K.; methodology, M.K.K.; software, S.-M.K. and M.K.K.; investigation, S.-M.K. and D.W.K.; validation, S.-M.K., D.W.K. and M.K.K.; formal analysis, S.-M.K., D.W.K. and M.K.K.; investigation, S.-M.K., D.W.K. and M.K.K.; writing—original draft preparation, S.-M.K. and D.W.K.; writing—review and editing, S.-M.K. and M.K.K.; visualization, S.-M.K. and D.W.K.; supervision, M.K.K.; project administration, M.K.K. All authors have read and agreed to the published version of the manuscript.

**Funding:** This research received no external funding.

**Institutional Review Board Statement:** Not applicable.

**Data Availability Statement:** No new data were created or analyzed in this study. Data sharing is not applicable to this article.

**Conflicts of Interest:** The authors declare no conflict of interest.

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
