# Peer review of "Design and Implementation of an Active Vibration Control Algorithm Using Servo Actuator Control Installed in Series with a Spring-Damper"

_applsci, doi:10.3390/app13053349_

Round 1

Reviewer 1 Report

This study proposes a system in which an actuator capable of accurately controlling displacement is connected in series with a support spring-damper. A new active vibration control algorithm for the proposed control system is also developed, which is termed the position input and position output. Numerical studies and experimental results show the efficacy of the proposed control system and the novel control algorithm for the vibration suppression of the lateral vibration of a vibration isolation table.

The issues studied in this paper are highly innovative, so I agree with its publication.

Author Response

Thanks for the comments.

Reviewer 2 Report

The submitted manuscript cannot be accepted for publication in this form, but it has a chance of acceptance after a major revision. My comments and suggestions are as follows:

1- Abstract gives information on the main feature of the performed study, but some details about the proposed system be added.

2- Authors must clarify necessity of the performed research. Aims and objectives of the study, and also differences with the previous review papers must be clearly mentioned.

3- The literature study must be enriched. For instance, authors must read and refer to the relevant papers: (a) https://doi.org/10.1016/j.apm.2013.10.039 (b) https://doi.org/10.1016/j.jsv.2022.117443 and other research works.

4- Authors must clearly emphasized the limitations and strengths of their study.

5- The current version of abstract is too short, but introduction is too long.

6- The main reference of each formula must be cited.

7- Why this particular technique is considered for this study? Scientific reason must be discussed.

8- All figures and curves must be illustrated in a high quality.

9- Reasons of large deviation must be discussed.

10- The curves must be illustrated in a more scientific way. For example, deviation (error bars) should be added. Also, there are sentences which have to be rewritten.

11- The conclusion must be more than just a summary of the manuscript. List of references must be updated based on the proposed papers. Please provide all changes by red color in the revised version.

Author Response

1- Abstract gives information on the main feature of the performed study, but some details about the proposed system be added.

Thanks for the comment. We added the following in abstract.

“These actuators generate force. In the case of a ball-screw mechanism device or a linear servomotor, it is an actuator that generates displacement. These actuators are represented as serial active systems. Serial active systems are structurally simpler than parallel active systems. However, there are very few studies on vibration isolation systems using serial active systems compared to parallel active systems. Since the two are different types of systems, a new control algorithm suitable for the serial active system is needed.”

“The proposed control algorithm uses the displacement of the system as an input and outputs the desired displacement of the actuator installed in series with the damper and spring. The proposed control algorithm increases the damping at the target frequency and reduces the response of the system. Numerical studies and experiments were conducted on the single-degree-of-freedom and multi-degree-of-freedom systems.”

2- Authors must clarify necessity of the performed research. Aims and objectives of the study, and also differences with the previous review papers must be clearly mentioned.

In this study, we are interested in the active control of lateral vibrations of the vibration isolation table. Most of the studies so far have used a parallel active system. On the other hand, research on vibration isolation system using a serial active system such as connecting a linear actuator and a spring-damper in series has not been done. Since it is a different type of system, a new control algorithm suitable for the serial active system is required. Therefore, in this study, a new algorithm for serial active type vibration isolation system was developed. We added the following in Introduction.

“The air spring and VCA described above are actuators that generate force. A ball-screw mechanism can be used to generate the desired displacement. However, displacement generating actuators cannot be installed parallel to spring dampers because the ball-screw mechanism acts as a lock when it is not working. In other words, displacement type actuator cannot be explained with parallel active type. So in this case, the case of connecting the linear actuator to the spring-damper in series has to be considered, as shown in Figure 1(d). The serial type shown in Figure 1(d) is structurally simpler than the parallel type shown in Figure 1(c). However, there are very few studies on vibration isolation systems using serial active systems compared to parallel active systems. Since the two are different types of systems, a new control algorithm suitable for the serial active system is needed. “

3- The literature study must be enriched. For instance, authors must read and refer to the relevant papers: (a) https://doi.org/10.1016/j.apm.2013.10.039 (b) https://doi.org/10.1016/j.jsv.2022.117443 and other research works.

Thanks for the comment. We added the following in introduction

“Vibration control is a crucial technology used in various structures such as build-ings, bridges, vehicles, and machinery to reduce unwanted vibrations and noise [1].”

“Devices such as tuned mass damper (TMD), tuned liquid damper (TLD), tuned liquid, column ball spring sliding damper (TLCBSSD), and tuned liquid column ball spring rolling damper (TLCBSRD) [3] can also be used.”

4- Authors must clearly emphasized the limitations and strengths of their study.

Thanks for the comment. We added the following in Discussion and Conclusions.

- The serial active type is structurally simpler than the parallel active type.

- However, compared to the parallel active system, the series active system must with-stand the static load when installed in the vertical direction.”

5- The current version of abstract is too short, but introduction is too long.

The journal instructed me to submit a large volume of 4000 words or more. Therefore, we tried to include a detailed explanation in the introduction part. More detailed information was added to the new abstract, and in the case of the introduction, the content was reorganized concisely, focusing on overlapping descriptions.

6- The main reference of each formula must be cited.

We added citation for Routh–Hurwitz stability criteria before equation (9).

  1. Routh, E.J. A Treatise on the Stability of a Given State of Motion, Particularly Steady Motion; Macmillan: 1877.

7- Why this particular technique is considered for this study? Scientific reason must be discussed.

Air spring and VCA, actuators commonly used in vibration isolation tables, is a type of actuator that generates force. These can be explained with parallel active type like Figure 1(c). Other actuator types generate displacement. These actuators form serial-type active systems. The serial type shown in Figure 1(d) is structurally simpler than the parallel type shown in Figure 1(c). However, there are very few studies on vibration isolation systems using serial-type active systems compared to parallel-type active systems. Since the two systems are totally different types of systems, a new control algorithm suitable for the serial active system is needed. We added related information at the abstract and end of the introduction.

8- All figures and curves must be illustrated in a high quality.

All graphs have been submitted at 600 dpi or higher quality. We changed the color of figure 4 for better visibility.

9- Reasons of large deviation must be discussed.

A graph showing the deviation does not exist in the paper. If deviation means the difference between numerical analysis and experimental results, the numerical analysis model and the experimental model are different cases, the two cannot be compared.

10- The curves must be illustrated in a more scientific way. For example, deviation (error bars) should be added. Also, there are sentences which have to be rewritten.

Since the algorithm is about a new algorithm that increases damping, an explanation of how the damping coefficient has changed has been added in the simulation and experimental results.

11- The conclusion must be more than just a summary of the manuscript.

We added the limitations and strengths of our study at Discussion and Conclusions.

Reviewer 3 Report

In this paper, the authors present a control proposal capable of suppressing the lateral vibrations of the vibration isolation table. Experimental results are presented to demonstrate the effectiveness of the proposed control.

The paper presents interesting and relevant results that justify its publication after minor corrections.

1- I suggest that authors include results comparing numerical results with experimental ones. 

Author Response

In this paper, the authors present a control proposal capable of suppressing the lateral vibrations of the vibration isolation table. Experimental results are presented to demonstrate the effectiveness of the proposed control.

The paper presents interesting and relevant results that justify its publication after minor corrections.

1- I suggest that authors include results comparing numerical results with experimental ones. 

Since the main purpose of this study is to study a new algorithm for increasing damping, an explanation of how the damping factors have changed has been added in the simulation and experimental results.

Reviewer 4 Report

The article presents an active vibration control system, starting from the design and finalizing with validation experiments. The material is well structured and the research methodology is adequate. The article is professionally written and could be interesting for readers, including for education purposes.

Author Response

Thanks for the comments.

Round 2

Reviewer 2 Report

The paper has been improved and corresponding modifications have been conducted. In my opinion, the current version can be considered for publication.